# Influence of Light Spectrum on Bread Wheat Head Colonization by *Fusarium graminearum* and on the Accumulation of Its Secondary Metabolites

**DOI:** 10.3390/plants14132013

**Published:** 2025-07-01

**Authors:** Minely Cerón-Bustamante, Francesco Tini, Giovanni Beccari, Andrea Onofri, Emilio Balducci, Michael Sulyok, Lorenzo Covarelli, Paolo Benincasa

**Affiliations:** 1Department of Agricultural, Food and Environmental Sciences, University of Perugia, 06121 Perugia, Italy; minelycb@gmail.com (M.C.-B.); francesco.tini@unipg.it (F.T.); giovanni.beccari@unipg.it (G.B.); andrea.onofri@unipg.it (A.O.); emilio.balducci@unipg.it (E.B.); paolo.benincasa@unipg.it (P.B.); 2Department of Agricultural Sciences, Institute of Bioanalytics and Agro-Metabolomics, BOKU University, A-3430 Tulln, Austria; michael.sulyok@boku.ac.at

**Keywords:** wheat, *Fusarium graminearum*, deoxynivalenol, deoxynivalenol-3-glucoside, light, trichothecenes

## Abstract

Previous studies indicated that light influences mycotoxin production and wheat’s defense responses to the cereal fungal pathogen *Fusarium graminearum*. Herein, the effect of different light wavelengths on *F. graminearum* colonization and secondary metabolite biosynthesis in bread wheat was assessed. Heads of a susceptible bread wheat cultivar were point-inoculated and exposed to red (627 nm), blue (470 nm), blue/red, and white light. Symptom severity, fungal DNA, and secondary metabolite accumulation were evaluated. Blue and red wavelengths reduced *F. graminearum* infection but had an opposite effect on the production of its fungal secondary metabolites. While blue light enhanced the accumulation of sesquiterpene mycotoxins, red light promoted the production of polyketide compounds. In addition, blue light stimulated deoxynivalenol glycosylation. These findings suggest that the light spectrum could affect mycotoxin contamination of wheat grains, highlighting the importance of light quality studies in field crops.

## 1. Introduction

Light is an important abiotic element that influences plant–pathogen interactions. The intensity, duration, and composition of light have a significant effect on the development of plant diseases. The use of light-emitting diodes (LEDs) that emit specific light wave bands has facilitated research on the effect of light quality on plant–pathogen interactions [1,2]. Particularly, LED lighting has permitted the characterization of light responses and the identification of light-responsive elements, thereby creating the potential for employing light wavelengths as a physical treatment to control plant diseases [3,4,5,6,7,8]. Several studies have proved the effectiveness of certain light wavelengths in disease management of controlled environment agriculture and postharvest handling [9,10,11,12]. However, there has been limited investigation into the impact of light composition on plant–pathogen interactions in open-field crops despite the extensive evidence demonstrating diurnal, seasonal, and geographical variations in the spectral quality of daylight [13,14,15]. Notably, some studies have improved our understanding of how light influences the growth, development, and virulence of certain fungal pathogens that affect cereal crops [16,17,18,19]. Particularly, the photoresponses *of Cercospora zeae-maydis*, *Magnaporthe oryzae*, *Zymoseptoria tritici*, and *Fusarium fujikuroi* have been extensively studied [17,19,20,21,22,23,24,25,26,27]. These studies have provided a comprehensive description of light responses, highlighting the importance of light quality and duration for fungal infection, colonization, as well as production of secondary metabolites. Nonetheless, a limited number of studies have been conducted on interaction with their respective plant hosts [28,29,30].

*Fusarium graminearum* is the main causal agent of Fusarium head blight (FHB) of bread wheat (*Triticum aestivum*), durum wheat (*Triticum turgidum* subsp. *durum*), and barley (*Hordeum vulgare*). *F. graminearum* infection causes premature bleaching of heads, floret abortion, or formation of white shriveled kernels [31]. In addition, due to the ability of *F. graminearum* to biosynthesize mycotoxins, grains can be contaminated with several mycotoxins, harmful to both humans and animals [32,33]. Some of the most important mycotoxins produced by *F. graminearum* are deoxynivalenol (DON), nivalenol (NIV), and zearalenone [34]. Both DON and NIV mycotoxins are part of the trichothecene family, a large group of tricyclic sesquiterpene compounds causing inhibitory effects in eukaryotic cells [35,36,37]. DON and its acetylated precursors [15 acetyl-deoxynivalenol (15ADON) and 3 acetyl-deoxynivalenol (3ADON)] are the main *Fusarium* mycotoxins found in raw materials and in grain-derived products [38,39,40,41]. In plants, DON has phytotoxic effects, causing bleaching of green tissues, such as leaves and heads, and alteration of seed germination, seedling growth, and root regeneration [42,43,44,45]. In addition, DON acts as a virulence factor that contributes to spreading the fungus within wheat heads [46,47].

Light has been demonstrated to affect the in vitro production of DON by *F. graminearum* [48,49]. In detail, LED illumination assays revealed that royal blue (455 nanometers) and blue (470 nanometers) light stimulated the biosynthesis of DON compared to red, yellow, and green wavelengths [49]. Conversely, it was reported that light had a detrimental effect on DON production compared to darkness [48]. These findings suggest that light affects the biosynthesis of DON and that the specific wavelengths involved play a crucial role in this response. In addition, a previous study demonstrated that light influences wheat defense response to *F. graminearum* and DON accumulation [42]. The authors of this study examined the expression of four well-established markers of plant defense response [*class III plant peroxidase* (*POX*), *phenylalanine ammonia-lyase* (*PAL*), *nonexpressor of pathogenesis-related genes-1* (*NPR1*), and *ß-1,3-glucanase* (*GLC*)] in wheat seedlings with varying levels of resistance to DON accumulation following exposure to light. The results revealed a significant 2.0–4.3-fold increase in the expression levels of *POX*, *PAL*, *NPR1*, and *GLC* compared to the coleoptiles of plants grown in the dark, suggesting that light could enhance the plant defense mechanisms against fungal infection and mycotoxin production.

Since light appears to be a crucial factor influencing *in vitro* DON production by *F. graminearum*, wheat’s defense response to *F. graminearum*, and DON *in planta* accumulation, we hypothesized that the light spectrum might have a differential effect on mycotoxin production and metabolism in wheat. Therefore, this study aims to assess the impact of four different light wavelengths (red, blue, blue/red, and white) on *F. graminearum* colonization and secondary metabolite accumulation in bread wheat heads. To the best of our knowledge, this study represents the first attempt to investigate the effect of the light spectrum on *F. graminearum* colonization and on the production of secondary metabolites in bread wheat heads.

## 2. Results

### 2.1. FHB Symptom Development in Bread Wheat Heads

Light spectra affected FHB symptom development in the bread wheat cultivar A416. After point inoculation with *F. graminearum* strain 65, FHB symptoms were evident upon a visual observation from 7 days post inoculation (dpi) under all light conditions. Even if FHB disease severity levels developed uniformly at 7 dpi, a significant effect of light treatment was observed at 14 and 21 dpi (Figure 1). In fact, from 14 dpi onwards, white and blue/red light significantly increased FHB symptom development compared to blue and red wavelengths (*p* < 0.05). At 21 dpi, exposure to blue and red light wavelengths led to a 66.7% and 61.5% reduction in FHB severity, respectively, compared to white light (*p* < 0.05). Interestingly, dichromatic blue/red light increased disease severity by 1.3-fold and 1.4-fold in comparison with blue and red light, respectively (*p* < 0.05). The control plants did not show any FHB symptoms.

### 2.2. F. graminearum Quantification in Bread Wheat Heads

The spectral light quality influenced *F. graminearum* DNA accumulation in wheat heads (Figure 2). At 60 dpi, *F. graminearum* DNA was detected in inoculated heads developed under all different light treatments. In general, a decreasing trend in fungal DNA was noticed from blue/red, white, and blue to red light. However, a significant difference was observed only between red and blue/red wavelengths (*p* < 0.05). The exposure to red light caused a 1.6-fold reduction in fungal DNA as compared to the blue/red wavelengths. *F. graminearum* DNA was not detected in the control plants.

### 2.3. F. graminearum Secondary Metabolite Accumulation in Bread Wheat Heads

Light conditions influenced the accumulation of secondary metabolites produced by *F. graminearum* in the bread wheat head. DON, its acetylated forms (3ADON and 15ADON), NIV, deoxynivalenol-3-glucoside (D3G), NIV-glucoside (NIVG), aurofusarin, zearalenone, culmorin, 5-hydroxyculmorin (5.HCUL), 15-hydroxyculmorin (15.HCUL), butenolide, sambucinol, chrysogine, and gibepyrone-D were detected by liquid chromatography–tandem mass spectrometry (LC-MS/MS) in all inoculated heads. To explore the variability and identify underlying patterns, a PCA was performed considering all secondary metabolites in all light conditions. The PCA confirmed the relationship between 15 secondary metabolites in response to all light conditions (Figure 3). The variance of the first component was mainly due to DON, 3ADON, sambucinol, 15.HCUL, 5.HCUL, NIV, 15ADON, chrysogine, and culmorin, which showed a similar response to blue and blue/red light. On the other hand, zearalenone, aurofusarin, and gibepyrone-D were grouped, showing higher production in the presence of red light. Interestingly, the location of zearalenone, aurofusarin, and gibepyrone-D in the dimensional space suggested a negative correlation with butenolide. On the contrary, zearalenone, aurofusarin, and gibepyrone-D seem to be unrelated to either trichothecenes or culmorin and hydroxy-culmorins (5.HCUL and 15.HCUL).

The ANOVA demonstrated a significant difference in the production of DON, NIV, D3G, aurofusarin, zearalenone, and gibepyrone-D in response to different light conditions (*p* < 0.01) (Figure 4).

Exposure to blue light significantly increased DON and NIV concentrations as compared to red wavelength (*p* < 0.05), while blue/red exposure resulted in intermediate levels of both mycotoxins. This additive effect was supported by a non-significant difference between red or blue light as compared to blue/red treatment. Interestingly, white and blue/red treatments led to similar DON and NIV concentrations. This suggests that the presence of both blue and red wavelengths was able to induce the production of both trichothecenes at similar levels to those observed under the full-spectrum white light. In addition, blue light also promoted DON glycosylation *in planta*. Remarkably, blue light favored the production of D3G by duplicating its concentration compared to other light conditions (*p* < 0.05) (Figure 4). Blue light not only stimulated DON and NIV biosynthesis but also the production of other trichothecenes. The PCA demonstrated that blue light favored the accumulation of the acetylated precursors 15ADON and 3ADON, confirming the favorable effect of this short wavelength on trichothecene accumulation. Similarly to trichothecenes, culmorin and its modified forms (5.HCUL 15.HCUL) accumulated largely under blue light. In contrast to what was observed for blue light, red light enhanced the production of the red pigment aurofusarin and the mycotoxin zearalenone. Red light exposure increased by 1.9-fold, 3.6-fold, and 4-fold aurofusarin concentration compared to blue, white, and blue/red light, respectively (*p* < 0.05). Similarly to aurofusarin, red wavelength significantly increased zearalenone concentration by 6.6-fold, 9.4-fold, and 11.2-fold in comparison to white, blue, and blue/red light, respectively (*p* < 0.05). No remarkable concentration difference was noticed among the other light wavelengths. In addition, a moderate, but still significant, increase in gibepyrone-D was observed in response to red light. Specifically, red light exposure significantly increased gibepyrone-D levels by 1.5-fold relative to the other light conditions (*p* < 0.05).

### 2.4. Correlation Between Fungal Biomass and Secondary Metabolites in Bread Wheat Heads

Considering that DON acts as a virulence factor facilitating *F. graminearum* colonization of wheat [46,47], a positive correlation between fungal amounts and DON concentration was expected in response to all light treatments. Indeed, a positive correlation was observed between fungal DNA and trichothecene levels in response to different lights (Figure 5). The amount of *F. graminearum* DNA significantly correlated with both DON and NIV concentrations under red (DON: r = 0.98, *p* < 0.001 | NIV: r = 0.95, *p* < 0.01) and blue/red (DON: r = 0.86, *p* < 0.05 | NIV: r = 0.84, *p* < 0.05) light. NIV also showed a significant association in blue light (NIV: r = 0.81, *p* < 0.05), while neither DON nor NIV demonstrated a significant relation with *F. graminearum* levels under white light (DON: r = 0.73 | NIV: r = 0.76). Interestingly, *F. graminearum* DNA and sambucinol also showed a positive and significant association under red (r = 0.98, *p* < 0.001) and blue/red (r = 0.89, *p* < 0.05) wavelengths. In conclusion, *F. graminearum* DNA positively correlated with DON concentration in all conditions, even if this correlation was not always significant. The reduction in *F. graminearum* amounts led to a lower DON concentration, and vice versa. A correlation analysis was performed to reveal any possible association between DON and D3G. A positive (r = 0.706) but non-significant correlation was observed between DON and D3G concentrations under blue light treatment. In contrast, DON and D3G showed a negative association under red (r = −0.684), blue/red (r = −0.82, *p* < 0.05), and white light conditions (r = −0.88, *p* < 0.05).

*F. graminearum* DNA and aurofusarin/zearalenone concentration exhibited a negative but non-significant correlation in all light treatments (Figure 4). An exception was observed under blue light, in which fungal biomass and zearalenone exhibited a significant association (r = −0.91, *p* < 0.05). Therefore, the high concentration of both aurofusarin and zearalenone under red light cannot be explained by an increase in the fungal biomass in planta. In contrast, *F. graminearum* quantities showed a significant positive correlation for the presence of gibepyrone-D in red (r = 0.97, *p* < 0.001) and white (r = 0.82, *p* < 0.05) light.

## 3. Discussion

It is well known that light affects the growth, reproduction, and virulence of fungal pathogens [16,50,51]. In *F. graminearum*, prior research has shown that light conditions influence the in vitro production of trichothecenes such as DON [48,49]. However, the present study provides the first direct evidence of how specific light wavelengths affect both colonization and mycotoxin production by *F. graminearum* in wheat heads under in planta conditions.

A central finding of this work is that monochromatic red and blue light significantly reduced the spread of *F. graminearum* compared to white light and a blue/red combination. This supports the hypothesis that light quality modulates fungal pathogenicity and plant defense responses. Previous studies have demonstrated that red and blue wavelengths influence pathogen development, host immunity, and virulence factor expression [12,29,30,52,53,54,55]. Blue light has been associated with reduced fungal growth but paradoxically enhanced virulence in some systems [49,55,56,57,58,59], while red light has been linked to suppression of infections via activation of defense pathways, such as systemic acquired resistance (SAR), in *Arabidopsis thaliana* and *Oryza sativa* [60,61]. Consistent with this, we previously demonstrated a fungistatic effect of red light on *Zymoseptoria tritici* in wheat, where lesion expansion and pycnidia formation were significantly inhibited under red light, whereas white light enhanced disease development [28]. These results support the broader conclusion that red light may enhance host resistance or interfere with critical stages of fungal colonization.

In *F. graminearum*, the reduced severity of FHB under blue light likely stems from a direct inhibitory effect on fungal growth, while red light appears to activate wheat defense mechanisms that limit pathogen spread. Interestingly, the combined blue/red treatment nullified these benefits, resulting in FHB severity comparable to white light. This suggests that the simultaneous presence of both wavelengths may counteract their individual inhibitory effects. A particularly novel discovery was the increase in DON accumulation under blue light, with levels 1.4-fold higher than those observed under red light. This finding aligns with earlier in vitro studies indicating that blue wavelengths stimulate DON biosynthesis [49]. Our results expand on these findings by confirming that blue light also enhances DON production in planta compared to red light, highlighting the complex interaction between light quality and mycotoxin biosynthesis in a natural plant–pathogen context.

Furthermore, blue light significantly elevated concentrations of deoxynivalenol-3-glucoside (D3G), with a two-fold increase relative to other light treatments. This may reflect the role of blue light in promoting *O*-glycosylation of organic compounds [62,63,64,65], potentially via upregulation of uridine diphosphate glycosyltransferases (UGTs). Although the total glycosylation efficiency remains uncertain, the observed increase in D3G even in a susceptible wheat genotype is noteworthy, given that reduced glycosylation is generally associated with heightened FHB susceptibility [45,66,67].

Together, these findings demonstrate that the light spectrum—particularly red and blue wavelengths—profoundly influences FHB progression, DON production, and its detoxification via glycosylation. These insights not only confirm the regulatory role of light in fungal pathogenicity and plant defense but also highlight the potential for using targeted light treatments to manage mycotoxin contamination. Nevertheless, further research is required to elucidate the molecular mechanisms underlying these responses, particularly the role of blue light in regulating glycosylation pathways across different wheat genotypes.

Besides DON, blue light also enhanced the accumulation of NIV, especially when compared to red light. PCA confirmed that blue light promotes the synthesis of acetylated DON precursors, such as 15ADON and 3ADON, indicating a more active trichothecene biosynthetic profile under this wavelength.

Additionally, blue light triggered a marked increase in culmorin and its hydroxylated derivatives (5.HCUL 15HCUL). As a tricyclic sesquiterpene diol synthesized from farnesyl pyrophosphate (FPP), culmorin is typically co-produced with trichothecenes in Fusarium-infected grain [40,68,69,70]. Given that culmorin can potentiate the phytotoxic effects of DON [71], the concurrent upregulation of both metabolite classes suggests that blue light may enhance early biosynthetic activity via FPP-derived intermediates. This opens the possibility that blue light acts upstream in secondary metabolism, stimulating multiple branches of toxin biosynthesis.

In contrast, red light preferentially enhanced the production of polyketide-derived metabolites, including aurofusarin, zearalenone, and gibepyrone-D. Aurofusarin is a red pigment produced by diverse *Fusarium* species possessing cytotoxic and teratogenic effects in human colon cells and embryos of poultry, respectively [72,73]. Notably, aurofusarin levels increased 3.6-fold under red light compared to white light. Unlike prior studies that found no effect of wavelength on aurofusarin under continuous light [48], our use of a 12 h light/dark cycle revealed significant induction, suggesting that both spectrum and photoperiod jointly regulate fungal secondary metabolism. Red light also stimulated zearalenone biosynthesis, a finding with toxicological relevance given the estrogenic properties of this mycotoxin [74]. Despite limited prior knowledge of environmental regulation of zearalenone, our results clearly demonstrate that red light can elevate its accumulation in planta. While some studies have reported an inverse relationship between aurofusarin and zearalenone [75], our data suggest that red light enables the co-production of both metabolites. A modest but significant increase in gibepyrone-D was also observed under red light. Gibepyrone-D, an αpyrone derived from gibepyrone-A via cytochrome P450 monooxygenases [76], is synthesized by a polyketide synthase (PKS) distinct from those producing DON or culmorin.

Interestingly, aurofusarin, zearalenone, and gibepyrone-D belong to one of the major classes of secondary metabolites in *Fusarium* species known as polyketides [77]. The biosynthetic pathways of such metabolites include steps catalyzed by multifunctional PKS enzymes. In summary, our data indicate that red light enhances the activity of PKS-mediated pathways, driving the biosynthesis of multiple secondary metabolites. Further transcriptomic analysis is needed to confirm whether red light upregulates specific PKS gene clusters, offering molecular insight into how the light spectrum influences fungal secondary metabolism *in planta*.

## 4. Materials and Methods

### 4.1. Fungal Strain and Inoculum Obtainment

*F. graminearum* strain 65, isolated from durum wheat grains in Umbria (central Italy), was used in this experiment because of its high DON/3ADON production [38]. The strain was grown on potato dextrose agar (PDA; Biolife Italiana, Milan, Italy) for seven days at 22 °C in the dark to produce mycelium. To obtain conidia, a mycelium plug (5 mm diameter) of a 7-day-old culture was placed in a mung bean broth prepared by boiling sterile water (1 L) and adding mung beans (40 g) for 10 min. Then, it was filtered and autoclaved. The inoculated liquid cultures were aerated with forced sterile air for 5 days at room temperature with a 10 h light and 14 h dark cycle. After 5 days, the conidial suspension was filtered through Miracloth (Millipore Corporation, Burlington, MA, USA). The conidia were collected using a 5804R centrifuge (Eppendorf, Hamburg, Germany) (3000 rpm for 15 min at 4 °C). Finally, their concentration was adjusted to 10^7^ conidia mL^−1^ with a hemocytometer. The inoculum was stored at −20 °C until it was used for plant inoculation.

### 4.2. Plant Material and Artificial Inoculation

Seeds of the bread wheat cultivar A416, highly susceptible to FHB, were surface-sterilized as described previously [78]. Successively, seeds were placed into 14 cm Petri dishes (Aptaca, Canelli, Italy) containing 3 filter papers (90 g, 15 mm diameter; Gruppo Cordenons, Milan, Italy) soaked with 15 mL of sterile water at 4 °C for 2 days in the dark and then incubated for 2 more days at 21 °C in the dark. Individual seedlings were transplanted in plastic pots (9 cm length, 9 cm width, 13 cm height) previously filled with peat (Klasmann-Deilmann GmbH, Geeste, Germany) and placed in a greenhouse at 20–25 °C with a 16 h photoperiod. At the full anthesis stage (BBCH 65), the main head of each plant was inoculated with 15 μL of a 10^7^ conidia mL^−^^1^ suspension of *F. graminearum* strain 65. In detail, point inoculation was performed by placing the suspension between the lemma and the palea of the two central spikelets of heads [79]. The inoculated heads were covered in clear plastic bags, previously sprayed with distilled water, and placed under white light for 48 h with a 12 h light and 12 h dark cycle. Non-inoculated heads were used as controls, covered in plastic bags, and exposed to the same light conditions. At 48 h post inoculation, bags were removed, and both inoculated and non-inoculated plants were placed under different light treatments. For each light condition, nine inoculated and nine non-inoculated wheat heads were used to ensure we had three heads for each of the three biological replicates. The entire experiment was repeated twice.

### 4.3. Light Treatments

Four different light treatments were applied, involving different light spectra. Blue (peak wavelength 470 nm), red (peak wavelength 627 nm), blue/red (50:50 ratio), and white light were applied using an LED lighting system (Figure 6). The LED system used has been described previously [80]. This system uses Philips Lumileds Luxeon Rebel LEDs, with each DSA3 lamp containing seven LED types that emit distinct wavelengths: royal blue (448 nm), blue (470 nm), cyan (505 nm), green (530 nm), amber (590 nm), red (627 nm), and deep red (655 nm). All four light spectra were administered with an equal cumulative photon flux density of 200 μmol photons m^−2^ s^−1^ under a 12 h light photoperiod.

### 4.4. Evaluation of FHB Symptoms

To determine the effect of light on FHB symptom development, the number of FHB symptomatic spikelets was evaluated at 7, 14, and 21 dpi. Disease severity was estimated by counting the number of symptomatic spikelets, expressed as a percentage (%), relative to the total number of spikelets per head [39,46]. Three scored heads were considered as one biological replicate. Three biological replicates per light treatment were used.

### 4.5. Fusarium graminearum Quantification

Entire wheat heads were sampled at 60 dpi from all treatments. Heads of both inoculated and non-inoculated plants were collected, frozen at −80 °C for 24 h, and freeze-dried for 24 h with a Heto PowerDry LL3000 (Thermo Fisher Scientific, Waltham, MA, USA). For each light treatment, the 3 whole heads forming one biological replicate were pooled together and ground with a Mixer Mill MM200 (Retsch, Dusseldorf, Germany) at 25 hertz for 3 min.

*F. graminearum* DNA amount was determined by quantitative real-time PCR (qPCR) assays. To set standard curves of both *F. graminearum* and wheat, DNA from pure fungal culture and healthy heads of the wheat cv. A416 was extracted.

The mycelium of a 7-day-old culture of the *F. graminearum* strain 65 and the heads of non-inoculated plants were frozen, freeze-dried, and ground by a Mixer Mill MM400 (Retsch). Finally, DNA was extracted as previously described [81]. The determination of DNA concentrations and the generation of *F. graminearum* and wheat standard curves were realized as described previously [82]. The limit of detection (LOD) of *F. graminearum* was 0.05 pg.

The DNA of the wheat heads was extracted using a previously mentioned method [39], and concentration was estimated and adjusted at 30 ng μL^−1^ as described previously [81]. The primers Fg16NF (5′-ACAGATGACAAGATTCAGGCACA 3′) and Fg16NR (5′ TTCTTTGACATCTGTTCAACCCA-3′) were used to quantify *F. graminearum* DNA [82]. The primers Hor1F (5′-TCTCTGGGTTTGAGGGTGAC-3′) and Hor2R (5′-GGCCCTTGTACCAGTCAAGGT-3′) were used to quantify wheat DNA [83]. The qPCR reactions we used were carried out as described in [81]. The PCR conditions consisted of 10 min at 95 °C, 45 cycles at 95 °C for 15 s and 61 °C for 1 min, heating at 95 °C for 10 s, cooling at 60 °C, and, finally, an increase to 95 °C at 0.5 °C every 5 s with the measurement of fluorescence. qPCR assays were realized in a CFX96 Real-Time System (Bio-Rad, Hercules, CA, USA). *F. graminearum* DNA amounts, calculated as the average of three replicates, were expressed as the fungal DNA (pg DNA) of each sample normalized to the bread wheat DNA (ng DNA).

### 4.6. Determination of F. graminearum Secondary Metabolites in Wheat Heads

One gram of each ground sample was extracted using 4 mL of extraction solvent (acetonitrile–water–acetic acid, 79:20:1, *v*/*v*/*v*) followed by a 1 + 1 dilution with acetonitrile–water–acetic acid, (20:79:1, *v*/*v*/*v*) and direct injection of 5 μL diluted extract. The chromatographic method, chromatographic and mass spectrometric parameters, quantification method, and results correction method we used have been described previously [84]. Briefly, target fungal metabolites were screened using LC-MS/MS with a QTrap 5500 system (Applied Biosystems, Thermo Fisher Scientific, Waltham, MA, USA) coupled to a Turbo Ion Spray source and an Agilent 1290 HPLC system. Separation was carried out at 25 °C on a Gemini^®^ C18 column (150 × 4.6 mm, 5 μm) with a matching C18 guard cartridge (Phenomenex, Torrance, CA, USA). Methodology and parameters followed [84]. Quantification was based on external calibration using serial dilutions of a multi-analyte stock, with results adjusted for apparent recoveries in wheat. Method accuracy was ensured through regular participation in proficiency testing.

### 4.7. Statistical Analysis

All statistical analyses were performed using the statistical software “R” (version 4.0; R Foundation for Statistical Computing, Vienna, Austria) (R Core Team, 2020). A preliminary descriptive data analysis was performed to explore the variability. Subsequently, the data of two independent experiments were pooled together in the same analysis and a linear model was fitted to the observed data to assess the effect of light on disease severity, fungal DNA amounts, and concentration of secondary metabolites. The light and year factors, together with the light by year interaction, were included as fixed effects. To meet the basic assumptions of normality and homoscedasticity, data about disease severity and fungal DNA amounts were square-root-transformed, while data about the concentration of secondary metabolites were log-transformed prior to the analyses. As the year and light by year effects were never significant, these effects were removed from the model to compare the means for the different light treatments.

For significant effects, a post hoc pairwise comparison was run in the transformed scale (square root or logarithm) with Tukey’s honestly significant difference (THSD) test (*p* < 0.05). For square-root-transformed data, the means and standard errors of the original data were displayed in figures, together with compact letter display based on the THSD test of the transformed data. For log-transformed data (concentrations of secondary metabolites), the means and standard errors were presented in the log-scale in order to avoid inconsistencies between the means of the original data and the results of compact letter display in the transformed scale. To visualize the data variability and relations among secondary metabolites, a PCA was performed on standardized concentrations of metabolites. The number of principal components was selected according to the Kaiser criterion, and only the factors displaying eigenvalues >1.00 were considered. The principal components were displayed on a ‘distance’ biplot. Finally, the ‘cor’ function was used to generate Pearson correlations between fungal DNA amounts and the significant fungal secondary metabolites per each type of wavelength. The ‘GGally’ and ‘ggplot2′ packages were used to produce correlation matrix heatmaps [85,86].

## 5. Conclusions

These findings demonstrate that both blue and red wavelengths influenced the colonization and secondary metabolism of *F. graminearum* during FHB infection in bread wheat (Figure 7). Both single wavelengths reduced the fungal spread in wheat heads, but they had the opposite effect on the production of some secondary metabolites. While blue light enhanced the accumulation of sesquiterpenes mycotoxins, red light promoted the production of polyketide compounds. This is evidence that light has a direct impact on the production of two of the most important mycotoxin classes in *F. graminearum*. These findings suggest that, even if some specific wavelengths interfere with host colonization, they can increase mycotoxin production, likely acting on photoreceptors or light-dependent genes involved in its biosynthetic pathway.

## Figures and Tables

**Figure 1 plants-14-02013-f001:**
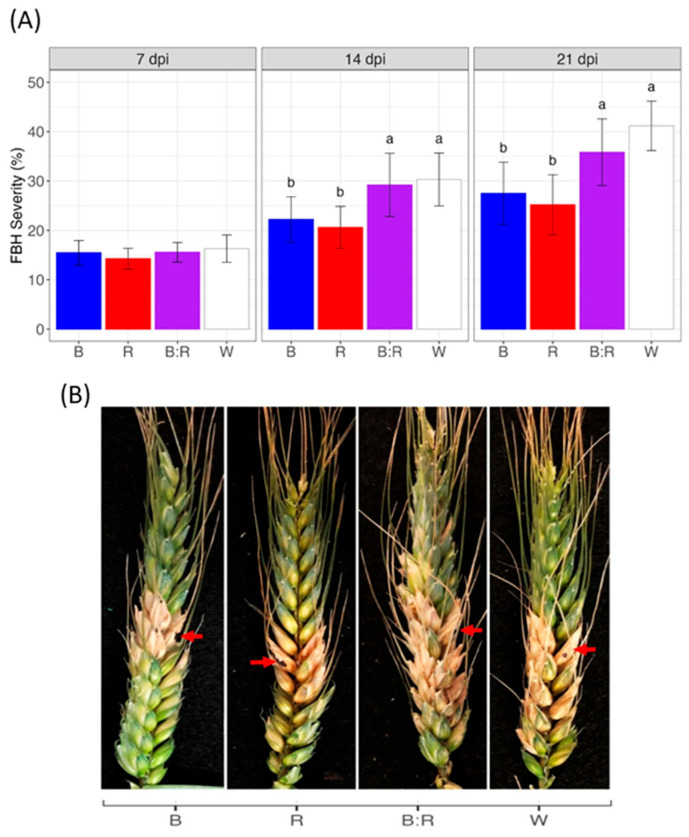
Effect of light spectra on Fusarium head blight (FHB) development in bread wheat. Heads were artificially inoculated with a conidial suspension (10^7^ conidia mL^−1^) of *F. graminearum* strain 56 and, after 48 h, incubated at 20 °C with exposure to blue (B), red (R), blue/red (B:R), and white (W) light. (**A**) Disease severity was evaluated at 7, 14, and 21 days post inoculation (dpi). Means with different letters are significantly different at *p* < 0.05, according to Tukey’s honestly significant difference (THSD) test on square root transformed data (to meet the basic assumptions for linear model fitting). Vertical bars indicate the average (±standard error) of the replicates. (**B**) FHB symptoms at 21 dpi in bread wheat plants under different light conditions. Red arrows indicate the inoculation point.

**Figure 2 plants-14-02013-f002:**
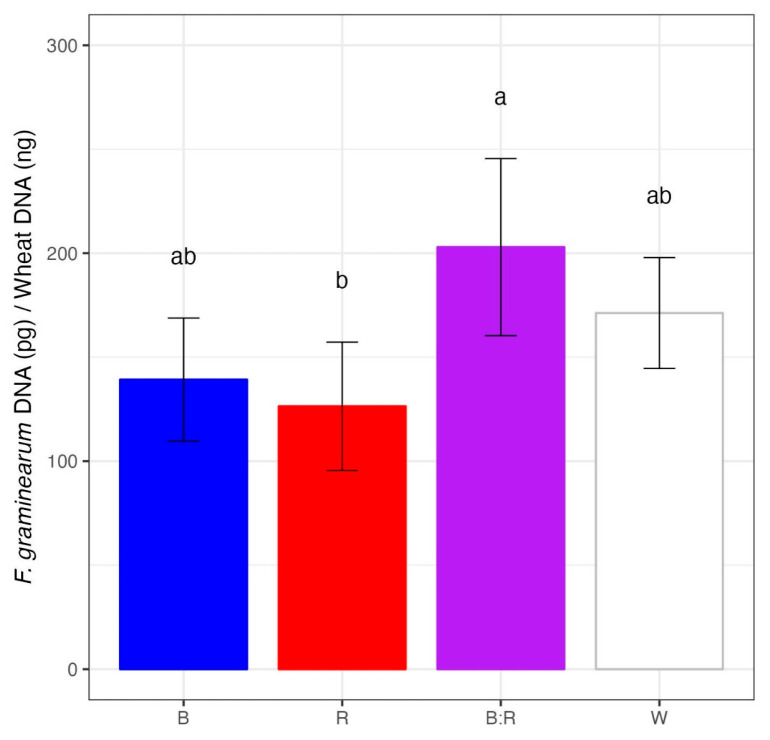
*F. graminearum* strain 65 DNA detected at 60 days post inoculation (dpi) in wheat heads grown under blue (R), red (R), blue/red (B:R), and white (W) light. After 60 dpi, inoculated heads were ground, and *F. graminearum* DNA was quantified by qPCR. The fungal DNA amount was expressed as the ratio of fungal DNA (pg) on wheat head DNA (ng). Means with different letters are significantly different at *p* < 0.05, according to Tukey’s honestly significant difference (THSD) test on square root transformed data (to meet the basic assumptions for linear model fitting). Vertical bars indicate the average (±standard error) of replicates.

**Figure 3 plants-14-02013-f003:**
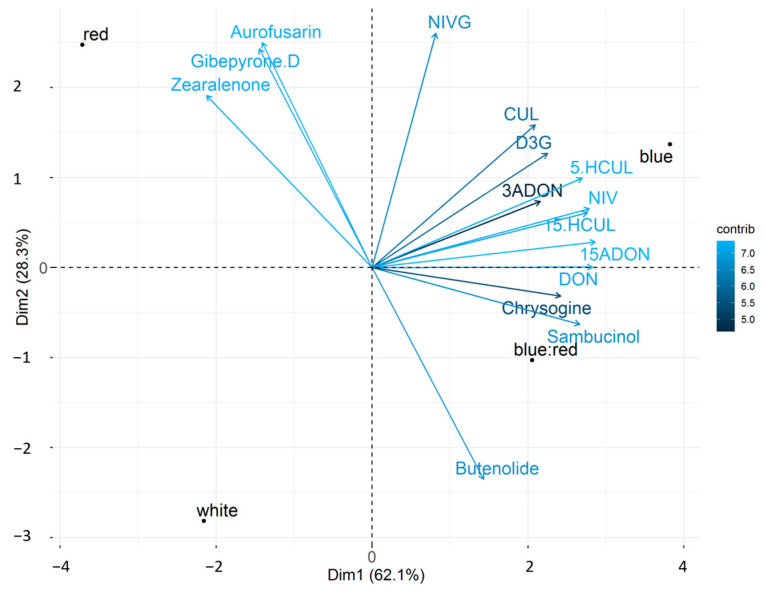
Secondary metabolite principal component analysis. The variability of 15 secondary metabolites (4 light conditions with 6 replicates each) is highlighted in a two-dimensional space. The variance explained by the first two components was 62.1% and 28.3%, respectively. The contribution of variables is perceived as a color gradient. Results for deoxynivalenol (DON), 3-acetildeoxynivalenol (3ADON), 15-acetildeoxynivalenol (15ADON), deoxynivalenol 3-glucoside (D3G), nivalenol (NIV), nivalenol glucoside (NIVG), culmorin (CUL), 5-hydroxyculmorin (5.HCUL), and 15-hydroxyculmorin (15.HCUL) are shown.

**Figure 4 plants-14-02013-f004:**
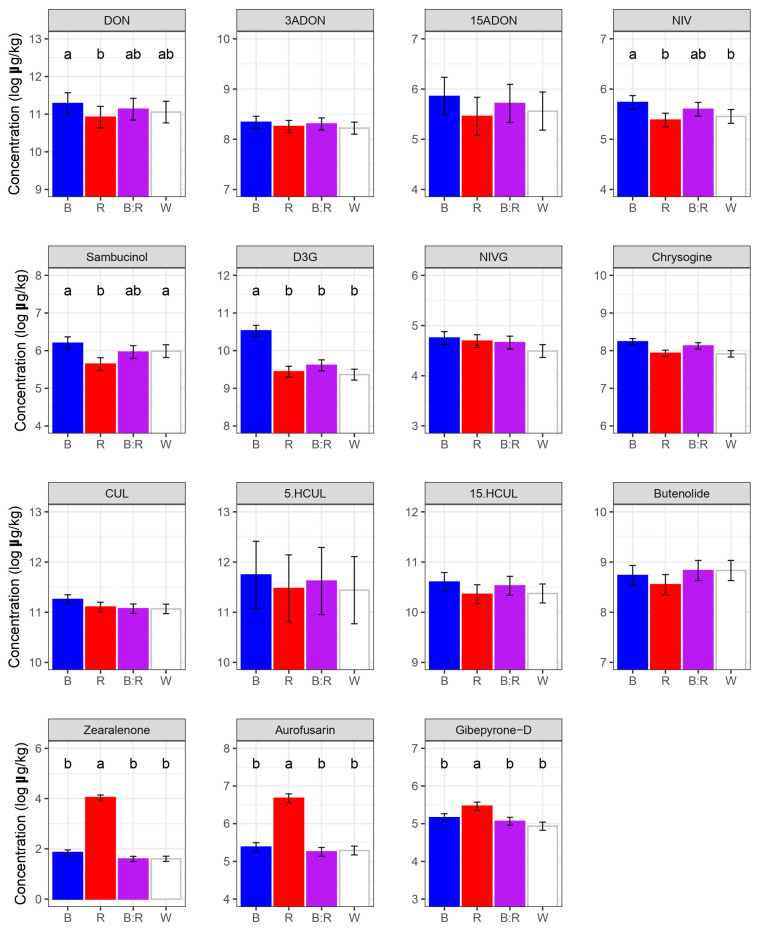
*F. graminearum* secondary metabolites detected in inoculated heads under blue (R), red (R), blue/red (B:R), and white (W) light. At 60 days post inoculation, inoculated heads were analyzed by LC-MS/MS. Results represent the average log-transformed concentration of *F. graminearum* secondary metabolites (±standard error of the difference), based on two independent experiments each including three replicates per treatment. Means with different letters are significantly different at *p* < 0.01, according to Tukey’s honestly significant difference (THSD) test on log-transformed data (to meet the basic assumptions for linear model fitting). Absence of letters indicates no differences among treatments. Results for deoxynivalenol (DON), 3-acetildeoxynivalenol (3ADON), 15-acetildeoxynivalenol (15ADON), deoxynivalenol 3-glucoside (D3G), nivalenol (NIV), nivalenol glucoside (NIVG), culmorin (CUL), 5-hydroxyculmorin (5.HCUL), and 15-hydroxyculmorin (15.HCUL) are shown.

**Figure 5 plants-14-02013-f005:**
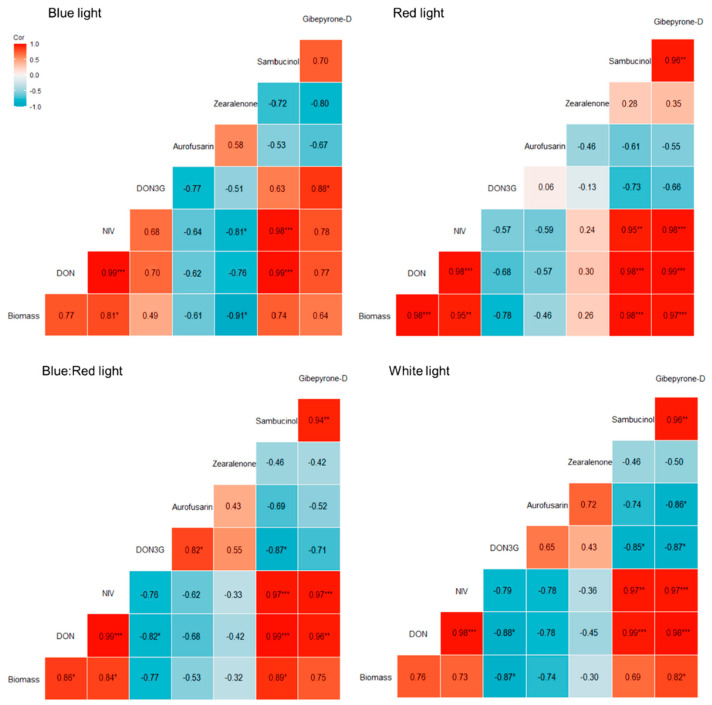
Correlation between fungal DNA and seven secondary metabolite concentrations in response to the four different light treatments applied. Pearson’s correlation coefficient analysis evaluated the relationship between *F. graminearum* DNA and mycotoxins that showed a differential production in response to light. Significance codes: *p* < 0.001 ‘***’, *p* < 0.01 ‘**’, and *p* < 0.05 ‘*’.

**Figure 6 plants-14-02013-f006:**
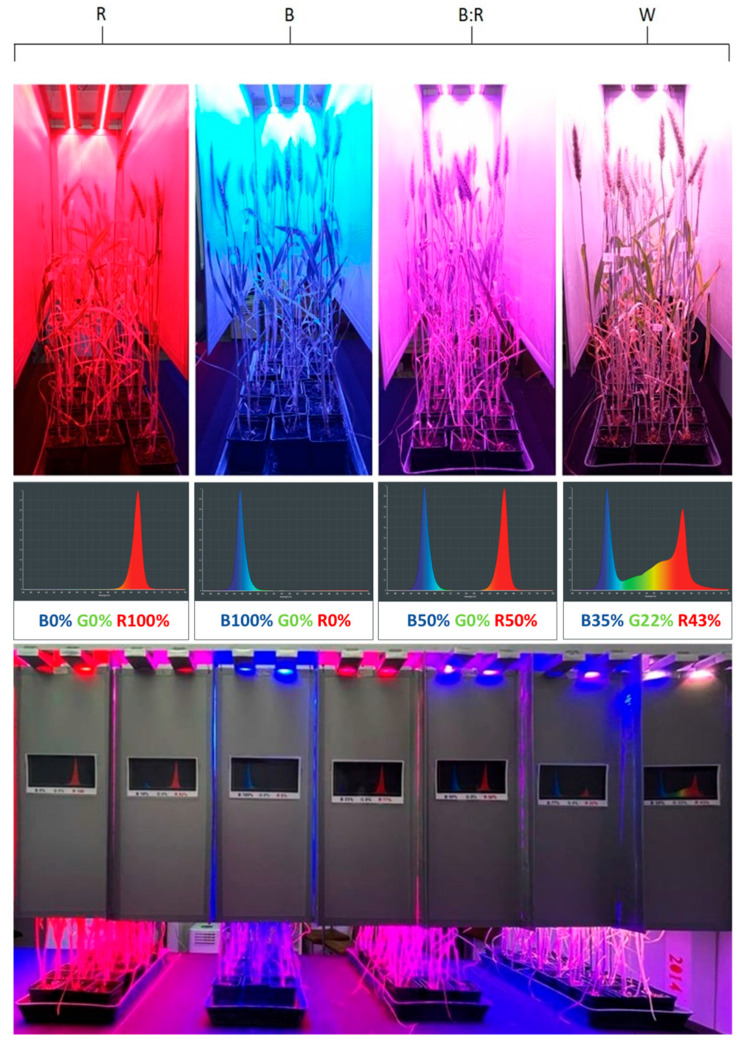
Light-emitting diode (LED) light chambers used in this study for the different light treatments. Blue (B) (λ peak at 470 nm), red (R) (λ peak at 627 nm), blue/red (B:R), and white (W) light were applied at the same cumulative photon flux density (200 μmol photons/m^−2^/s) with a photoperiod of 12 h.

**Figure 7 plants-14-02013-f007:**
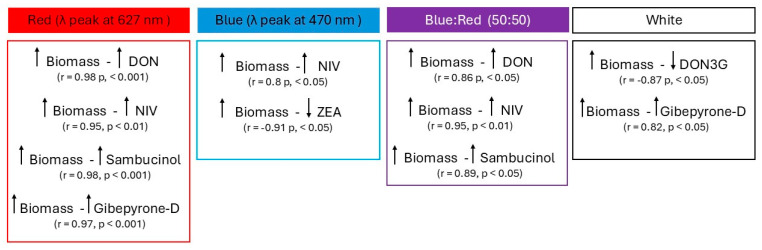
Conclusive model summarizing the relationship between fungal biomass and secondary metabolite production in response to the four different light treatments applied.

## Data Availability

The data presented in this study are available on request from the corresponding author. The data belong to the University of Perugia.

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
