# Peer review of "Influence of Light Spectrum on Bread Wheat Head Colonization by Fusarium graminearum and on the Accumulation of Its Secondary Metabolites"

_plants, 2025, doi:10.3390/plants14132013_

Round 1
Reviewer 1 Report
Comments and Suggestions for Authors
-
How did you measure biomass, as mentioned in Section 2.3 (Correlation between fungal biomass and secondary metabolites in bread wheat heads)?
-
Please expand the methodology described in Section 4.3, providing more detailed information.
-
Please elaborate on the methodology in Section 4.6, including specific details about the methods used, instruments employed, the list of secondary metabolites analyzed, the quantification procedure, etc.
-
Section 4.7 is titled "Determination of F. graminearum secondary metabolites in wheat heads," but only statistical methods are described. Please revise the heading or provide the appropriate methodological details.
- A conclusive model summarizing the relationship between fungal biomass and secondary metabolite production would be appropriate.
Author Response
We are grateful to reviewer 1 for the valuable revision that certainly improves the quality of the paper. Please find below the response to each of the reviewer’s comments, which we have tried to address in the manuscript to meet the requirements.
- How did you measure biomass, as mentioned in Section 2.3 (Correlation between fungal biomass and secondary metabolites in bread wheat heads)?
Biomass quantification was performed as described in section 4.5. We have corrected the section title 4.5 to clarify this query. Please see line 498 of the revised manuscript.
- Please expand the methodology described in Section 4.3, providing more detailed information.
Thanks for this comment. Section 4.3 has been completed with more information. Please see lines 480-485 of the revised manuscript.
- Please elaborate on the methodology in Section 4.6, including specific details about the methods used, instruments employed, the list of secondary metabolites analyzed, the quantification procedure, etc.
Section 4.6 has been updated to include additional details about the methodology, including the methods used, the instruments employed, and the quantification procedure. Please see lines 531-538 of the revised manuscript.
- Section 4.7 is titled "Determination of graminearum secondary metabolites in wheat heads," but only statistical methods are described. Please revise the heading or provide the appropriate methodological details.
Thanks for this comment. The title of Section 4.7 has been corrected to reflect the statistical methods described in the section accurately. Please see line 541 of the revised manuscript.
- A conclusive model summarizing the relationship between fungal biomass and secondary metabolite production would be appropriate.
We appreciate the reviewer’s observation. An image summarizing the relationship between fungal biomass and mycotoxin concentrations has been added to facilitate easy interpretation of the results. Please see lines 589-591 of the revised manuscript.
Reviewer 2 Report
Comments and Suggestions for Authors
Comments to authors
The manuscript ID: plants-3633304 entitled ‘’Influence of Light Spectrum on Bread Wheat Head Colonization By Fusarium graminearum and on the Accumulation of its Secondary Metabolite’’ by Cerón-Bustamante et al. investigated the effect of different light wavelengths on F. graminearum colonization and secondary metabolite biosynthesis in bread wheat, highlighting the importance of light quality studies in field crops. I consider the idea and intention of the article to be good, however, some weakness need to be considered before acceptance to publication. Below you can find to be considered the most relevant:
Major points:
- Why the authors examined specifically the expression of these four defense genes (POX, PAL, NPR1, and GLC) in wheat seedlings following exposure to light?
- Why the authors choose these three-point inoculation time (7, 14 and 21 dpi).
- The authors should specify the ratio of blue:red light and explain the choice of this ratio.
- The discussion section should be improved by demonstrating the importance of data obtained.
Minor points:
- I suggest to added the keyword ‘’ Fusarium graminearum’’ and change ‘’cereals’’ by ‘’wheat’’.
- Line 64 and 79, ‘’in vitro’’ should be written in italic. Please check the whole manuscript.
- Line 97, define the abbreviation ‘’dpi’’ (days post inoculation) in its first appearance in the manuscript.
- The quality of figure 1A need to be improved
- Lines 109, 335 and 343, ‘’20° C’’ should be changed to ‘’20°C’’. Please check the whole manuscript
- ‘’F. graminearum’’ should be written in italic. Please check the whole manuscript.
- Figure 2 should not be so big. Please use the same size for all histograms.
- Please specify the meaning of ‘’ Tukey HSD’’ statistical test.
- Line 138, change the word ‘’ evidenced’’ by ‘’confirmed’’.
- Line 300-302, revise this sentence ‘’ However, it is important to specify that F. graminearum colonies were exposed to continuous light while [48], while we analyzed the production of au rofusarin by F. graminearum in planta under 12 hours light cycle’’.
- Line 339, change ‘’hours’’ by ‘’h’’. Please check the whole manuscript.
- Some grammatical and punctuations in the text of manuscript. Please check.
- ‘’Conflicts of Interest’’ should be a new section.
- Abbreviations section should be presented in the beginning of the manuscript. Please refer to the following website for more information: https://www.mdpi.com/authors/references.
Author Response
We are grateful to reviewer 2 for the valuable revision that certainly improves the quality of the paper. Please find below the response to each of the reviewer’s comments, which we have tried to address in the manuscript to meet the requirements.
- Why the authors examined specifically the expression of these four defense genes (POX, PAL, NPR1, and GLC) in wheat seedlings following exposure to light?
Thanks for this comment. The authors we cited in the introduction section specifically examined the expression of POX, PAL, NPR1, and GLC because these genes are well-established markers of plant defense responses and play key roles in different defense pathways. We have made a minor modification to the paragraph. Please see lines 99-103 of the revised manuscript.
- Why the authors choose these three-point inoculation time (7, 14 and 21 dpi).
The three-time points we have selected refer to days post inoculation (dpi) in order to monitor the progression of Fusarium head blight (FHB) symptoms. FHB is typically evaluated at 7, 14, and 21 days post-inoculation (dpi) to monitor the progression of disease development over time and to capture key stages of host-pathogen interaction.
- 7 dpi represents the early phase of infection when initial symptoms and fungal colonization begin to appear.
- 14 dpi reflects the active phase of disease progression, including visible symptom development and potential spread within the spikelets.
- 21 dpi allows for assessment of the full extent of disease severity and the cumulative impact on the spike, including tissue necrosis and mycotoxin (e.g., DON) accumulation.
- The authors should specify the ratio of blue:red light and explain the choice of this ratio.
The blue:red light ratio was 50:50, meaning that half of the lamps emitted blue light and the other half emitted red light. We selected the blue:red wavelengths because both are primary light spectra perceived by plant photoreceptors, specifically, phototropins and cryptochromes respond to blue light, while phytochromes are sensitive to red light (van Leperen, 2012 doi: 10.17660/actahortic.2012.956.12.). Moreover, the combination of blue and red light has been shown to synergistically influence plant growth and development more effectively than either wavelength alone, as these wavelengths regulate complementary physiological processes such as photosynthesis, morphology, and secondary metabolite production (Bartucca et al., 2020 doi: 10.1021/acs.jafc.0c03851). Since we had already investigated the effects of individual wavelengths, we wanted to examine the combined effect of both wavelengths and compare it to the effects of each wavelength individually. This information has been included in the revised manuscript. Please see line 478 of the revised manuscript.
- The discussion section should be improved by demonstrating the importance of data obtained.
We appreciate the reviewer’s observation. The discussion section has been thoroughly revised to place emphasis on the experimental results. Key findings are now clearly highlighted and their significance in the context of existing literature is more explicitly discussed. Please see lines 314-397 of the revised manuscript.
- I suggest to added the keyword ‘’Fusarium graminearum’’ and change ‘’cereals’’ by ‘’wheat’’.
The correction has been made as requested. Please see line 25 of the revised manuscript.
- Line 64 and 79, ‘’in vitro’’ should be written in italic. Please check the whole manuscript.
The correction has been made as requested.
- Line 97, define the abbreviation ‘’dpi’’ (days post inoculation) in its first appearance in the manuscript.
The correction has been made as requested. Please see line 126 of the revised manuscript.
- The quality of figure 1A need to be improved
We improved the quality of figure 1A. Please see line 138 of the revised manuscript.
- Lines 109, 335 and 343, ‘’20° C’’ should be changed to ‘’20°C’’. Please check the whole manuscript.
The correction has been made as requested.
- ‘’F. graminearum’’ should be written in italic. Please check the whole manuscript.
The correction has been made as requested.
- Figure 2 should not be so big. Please use the same size for all histograms.
The correction has been made as requested. Please see line 156 of the revised manuscript.
- Please specify the meaning of ‘’ Tukey HSD’’ statistical test.
The correction has been made as requested. Please see line 161 of the revised manuscript.
- Line 138, change the word ‘’ evidenced’’ by ‘’confirmed’’.
The correction has been made as requested. Please see lines 172-173 of the revised manuscript.
- Line 300-302, revise this sentence “However, it is important to specify that F. graminearum colonies were exposed to continuous light while [48], while we analyzed the production of aurofusarin by F. graminearum in planta under 12 hours light cycle’’.
We have enhanced the discussion section, including that part.
- Line 339, change ‘’hours’’ by ‘’h’’. Please check the whole manuscript.
The correction has been made as requested. Please see line 451.
- Some grammatical and punctuations in the text of manuscript. Please check.
We reviewed the grammar of this manuscript.
- ‘’Conflicts of Interest’’ should be a new section.
We have now provided the Conflicts of Interest section. Please see line 604 of the revised manuscript.
- Abbreviations section should be presented in the beginning of the manuscript. Please refer to the following website for more information: https://www.mdpi.com/authors/references.
We have included a list of abbreviations immediately after the abstract in addition to the list of abbreviations already provided before the references section. Please see lines 28-54 of the revised manuscript.
Reviewer 3 Report
Comments and Suggestions for Authors
Light significantly influences plant growth and development. This manuscript investigates the effects of red light, blue light, and their combined treatment on Fusarium graminearum infection and metabolite profiles in wheat. The results demonstrate that both monochromatic wavelengths reduce Fusarium graminearum spread in wheat spikes, with differential impacts on metabolite accumulation. This study provides evidence for spectral regulation of mycotoxin metabolism in wheat, laying a foundation for future mycotoxin management strategies.
- Line 101: Please clarify how the 1.6-fold and 1.5-fold reductions were calculated; it is recommended to express these as percentage decreases.
- Figure 1B visually indicates more severe Fusarium head blight (FHB) symptoms in bread wheat under BR treatment at 21 days compared to W treatment. If possible, replace the image to better align with the results shown in Figure 1A.
- The significance annotation for sambucinol in Figure 4 is incorrect.
- The dataset presented is limited. Although the authors performed secondary analyses (PCA and correlation analysis), additional experiments are necessary to robustly demonstrate the effects of light spectrum on wheat FHB infection and metabolite content. The current manuscript lacks sufficient data to qualify as a research article in Plants.
- Line 158: Please confirm whether the significance threshold is p = 0.01 or p = 0.05 as indicated in Figure 4.
- Line 205: Remove “DON: r = 0.77”.
- Line 206: Fusarium graminearum should be italicized.
- Line 254: Revise the sentence “The results obtained in this study showed also that the accumulation of DON in wheat heads can be stimulated by blue light, while it can be repressed under red light.” Is the control treatment white light? There is no significant difference between red or blue light treatments compared to white light.
- The discussion is thorough; however, 101 references are excessive for a research article. It is advisable to increase experimental data to reduce reliance on speculative discussion and inference.
- Please specify the ratio of red to blue light used. What is the rationale for selecting this particular red:blue light ratio?
- Strengthen the comparison and discussion with the author's previous research, particularly doi.org/10.3390/jof9060670.
Author Response
We are grateful to reviewer 3 for the valuable revision that certainly improves the quality of the paper. Please find below the response to each of the reviewer’s comments, which we have tried to address in the manuscript to meet the requirements.
- Line 101: Please clarify how the 1.6-fold and 1.5-fold reductions were calculated; it is recommended to express these as percentage decreases.
Reductions have been expressed as percentages. Please see lines 130-132 of the revised version of the manuscript.
- Figure 1B visually indicates more severe Fusarium head blight (FHB) symptoms in bread wheat under BR treatment at 21 days compared to W treatment. If possible, replace the image to better align with the results shown in Figure 1A.
We appreciate the reviewer’s observation. The number of infected spikelets was slightly higher in the plants grown under white light. In Figure 1A, 8 spikelets were counted under blue:red light and 9 spikelets under white light. Although there was a slightly higher incidence under white light, statistical analysis did not reveal a significant difference compared to the blue:red light treatment. Unfortunately, we do not have better images than the ones presented.
- The significance annotation for sambucinol in Figure 4 is incorrect.
Thanks for this comment. For log-transformed data (concentrations of secondary metabolites), in the revised version of the manuscript, the means and standard errors were presented in the log scale, to avoid inconsistencies between the means of the original data and the results of compact letter display in the transformed scale. Please see lines 194-205 and of 554-561 of the revised manuscript.
- The dataset presented is limited. Although the authors performed secondary analyses (PCA and correlation analysis), additional experiments are necessary to robustly demonstrate the effects of light spectrum on wheat FHB infection and metabolite content. The current manuscript lacks sufficient data to qualify as a research article in Plants.
We thank the reviewer for pointing out a possible weakness of our work, and we understand the suggestion to perform additional experiments. However, we would like to highlight that, thanks also to the helpful comments of the Reviewers, the revised version of the manuscript gives interesting information and data for future studies in this field. For this reason, and without claiming to explain more than what we have observed, we kindly ask the reviewer to reconsider the revised manuscript and evaluate it for possible publication in Plants.
- Line 158: Please confirm whether the significance threshold is p = 0.01 or p = 0.05 as indicated in Figure 4.
We have corrected the p value in footnote of Figure 4.
- Line 205: Remove “DON: r = 0.77”.
This correction has been made as requested. Please see line 249 of the revised manuscript.
- Line 206: Fusarium graminearum should be italicized.
This correction has been made as requested.
- Line 254: Revise the sentence “The results obtained in this study showed also that the accumulation of DON in wheat heads can be stimulated by blue light, while it can be repressed under red light.” Is the control treatment white light? There is no significant difference between red or blue light treatments compared to white light.
We have revised the discussion section. Please see the revised manuscript.
- The discussion is thorough; however, 101 references are excessive for a research article. It is advisable to increase experimental data to reduce reliance on speculative discussion and inference.
We appreciate the reviewer’s observation. We have removed some references without compromising the content. Each citation has been carefully selected to support key statements and to contextualize our findings within the existing body of literature.
- Please specify the ratio of red to blue light used. What is the rationale for selecting this particular red:blue light ratio?
Thanks for this comment. As already explained also to reviewer 2 the blue:red light ratio was 50:50, meaning that half of the lamps emitted blue light and the other half emitted red light. We selected the blue:red wavelengths because both are primary light spectra perceived by plant photoreceptors, specifically, phototropins and cryptochromes respond to blue light, while phytochromes are sensitive to red light (van Leperen, 2012 doi: 10.17660/actahortic.2012.956.12.). Moreover, the combination of blue and red light has been shown to synergistically influence plant growth and development more effectively than either wavelength alone, as these wavelengths regulate complementary physiological processes such as photosynthesis, morphology, and secondary metabolite production (Bartucca et al., 2020 doi: 10.1021/acs.jafc.0c03851). Since we had already investigated the effects of individual wavelengths, we wanted to examine the combined effect of both wavelengths and compare it to the effects of each wavelength individually. This information has been included in the revised manuscript. Please see lines 477-485 of the revised manuscript.
- Strengthen the comparison and discussion with the author's previous research, particularly doi.org/10.3390/jof9060670.
We appreciate the reviewer’s observation. We have compared the results obtained in this study with previous research and have added the relevant reference in the revised discussion. Please see lines 328-333 of the revised manuscript.
Round 2
Reviewer 1 Report
Comments and Suggestions for Authors
The authors have significantly improved the manuscript. It should be accepted in its present form.
Reviewer 2 Report
Comments and Suggestions for Authors
The paper is well improved and the authors were clearly responded to all queries.
I still have a one more minor point to be verified by the journal instruction: Abbreviations should be presented one time in the manuscript, (before the Introduction or before References section).
Reviewer 3 Report
Comments and Suggestions for Authors
Accept in present form.